# Emerging Trends in Active Packaging for Food: A Six-Year Review

**DOI:** 10.3390/foods14152713

**Published:** 2025-08-01

**Authors:** Mariana A. Andrade, Cássia H. Barbosa, Regiane Ribeiro-Santos, Sidney Tomé, Ana Luísa Fernando, Ana Sanches Silva, Fernanda Vilarinho

**Affiliations:** 1Department of Food and Nutrition, National Institute of Health Doutor Ricardo Jorge, Av. Padre Cruz, 1649-016 Lisbon, Portugal; mariana.andrade@insa.min-saude.pt (M.A.A.); cassia.barbosa@insa.min-saude.pt (C.H.B.); sidney.tome@insa.min-saude.pt (S.T.); 2REQUIMTE/LAQV, R. D. Manuel II, Apartado, 55142 Oporto, Portugal; 3Mechanical Engineering and Resource Sustainability Center (METRICS), Department of Chemistry, NOVA School of Science and Technology, Campus de Caparica, NOVA University Lisbon, 2829-516 Caparica, Portugal; ala@fct.unl.pt; 4Federal Institute of Rondônia, Ariquemes CEP 76.878-899, RO, Brazil; ribeirorsantos@gmail.com; 5Faculty of Pharmacy, University of Coimbra, Azinhaga de Santa Comba, 3000-548 Coimbra, Portugal; asanchessilva@ff.uc.pt; 6Centro de Estudos de Ciências Animal (CECA), ICETA, Universidade do Porto, 4050-313 Oporto, Portugal; 7Associate Laboratory for Animal and Veterinary Sciences (AL4AnimalS), 1300-477 Lisboa, Portugal

**Keywords:** active food packaging, essential oils, natural extracts, phenolic compounds, biodegradable polymers, encapsulation, circular economy, food preservation

## Abstract

The development of active food packaging has evolved rapidly in recent years, offering innovative solutions to enhance food preservation and safety while addressing sustainability challenges. This review compiles and analyzes recent advancements (2019–2024) in release-type active packaging, focusing on essential oils, natural extracts, and phenolic compounds as active agents. Primarily plant-derived, these compounds exhibit significant antioxidant and antimicrobial activities, extending shelf life and enhancing food quality. Technological strategies such as encapsulation and polymer blending have been increasingly adopted to overcome challenges related to volatility, solubility, and sensory impact. Integrating bio-based polymers, including chitosan, starch, and polylactic acid, further supports the development of environmentally friendly packaging systems. This review also highlights trends in compound-specific research, release mechanisms, and commercial applications, including a detailed analysis of patents and case studies across various food matrices. These developments have already been translated into practical applications, such as antimicrobial sachets for meat and essential oil-based pads for fresh produce. Moreover, by promoting the valorization of agro-industrial by-products and the use of biodegradable materials, emission-type active packaging contributes to the principles of the circular economy. This comprehensive overview underscores the potential of natural bioactive compounds in advancing sustainable and functional food packaging technologies.

## 1. Introduction

As consumers, we have at our disposal a vast array of packaging. Packaging systems are diverse and multifunctional, from simple paper packaging to complex multi-layer packaging made of various types of plastic with multiple functions, including the basic single-layer transparent plastic packaging. Regarding food packaging, the options are also wide, and the primary function is to protect food during transport, distribution, and storage until it reaches the end consumer.

Throughout the years, several food packaging concepts have emerged, including active food packaging. This concept is based on the interaction between the packaging and the packaged food, with the goal to extend the food’s shelf-life or to maintain or improve the condition of the packaged food [1]. Active-releasing systems are designed to deliberately release substances, such as antioxidants, antimicrobials, carbon dioxide, flavors, ethylene, and ethanol, from the packaging into the food or the environment surrounding the food [2]. This can be an advantage, since the additive is incorporated into the packaging matrix rather than directly into the food, potentially reducing the amount of additives consumed by the final consumer [2]. For the purposes of this review, only this type of system will be considered.

The Regulation EC 1333/2008 [3] and its amendments set the definitions and conditions of use of food additives. For the direct and indirect use of additives in foods, the substance in question must be authorized and listed in this Regulation. Several studies have demonstrated that long-term exposure to synthetic food additives can have harmful effects on human health, leading to the development of diabetes, liver and kidney damage, infertility, allergic reactions, and various types of cancer [4]. As a result, the industry’s interest has shifted to look for natural-based additives with similar activities/functions to the synthetic ones. Among these natural-based additives, essential oils and phenolic compounds have gained particular attention due to their well-documented antioxidant and antimicrobial properties [5,6,7]. However, these natural-based additives, although very promising, still present some technological drawbacks, such as high volatilization, low solubility, chemical instability, and hydrophobicity [4]. One way to address these difficulties is through active food packaging techniques, since the additive is indirectly applied to the food.

In the category of natural compounds, phenolic compounds form a highly diverse group that has gained increasing attention over the years. These secondary metabolites produced by plants are the second most abundant group in the plant kingdom and are responsible for several characteristics, namely defense against ultraviolet radiation, biotic and abiotic stress, pathogens, and herbivores, and they are also responsible for quality attributes of vegetables and fruits such as flavor and color [8]. Generally, these compounds are obtained in the form of extracts or essential oils, which can be extracted by several methods, such as solvent extraction and microwave-assisted extraction, among others.

Essential oils (EOs) have been used by human civilization since ancient times for multiple purposes, such as religious rituals, fragrances, and medicinal purposes [9]. EOs are highly volatile substances that can be isolated through steam or dry distillation or by mechanical processes. Exhibiting low solubility in water, these substances dissolve well in most organic solvents and blend easily with vegetable oils, fats, and waxes [10]. Regarding its general chemical composition, the majority of EOs are composed mainly of hydrocarbons (mono- and sesquiterpenes), such as limonene and pinene, or oxygenated compounds, such as eugenol, cinnamaldehyde, and linalool. However, in some cases, such as thyme and peppermint, the predominant derivatives are aromatic, such as thymol and carvacrol [10].

Unlike EOs, plant extracts can have non-volatile compounds, such as tannins, flavonoids, vitamins, and minerals. These non-volatile compounds, with high molecular weight, can be extracted by solvents (solid–liquid extraction), maceration, or supercritical fluids [11,12,13]. These EOs and natural extracts have powerful biological activities due to their complex chemical composition. While hydrocarbons do not have direct biological activity, they can potentiate the effects of oxygenated and phenolic compounds, which are mainly responsible for EOs and extracts’ biological activities.

With more than 8000 identified phenolic structures, phenolic compounds are secondary metabolites produced by plants for their natural defense against UV and IR radiation, pathogens, and predators. Characterized by having at least one aromatic ring with one or more OH groups, they can be divided into simple (one aromatic ring) and polyphenol (more than one aromatic ring) classes [14,15]. The polyphenols class can be divided into flavonoids and non-flavonoids, which comprise tannins, lignans, stilbenes, and phenolic acids [15,16].

Research on active food packaging has increased significantly over the last 20 years, with exponential growth in the past 5 years (Table 1). A simple search in the PubMed database from 2004 to 2024 (publication year), using only the keywords ‘active food packaging,’ returned 6320 results. Of these, 4428 were published between 2019 and 2024, with 2020 results from 2023 and 2024. According to the PubMed search, the journal with the most publications is the *International Journal of Biological Macromolecules* (1045 papers), followed by *Food Chemistry* (379 papers) and *Polymers* (347 papers).

Using the Web of Science database with the same keywords, the search yielded 5740 results from 2004 to 2024 (publication year) (Table 1). Of these, 4191 were published between 2019 and 2024, and 1968 were from 2023 and 2024. According to the Web of Science search, between 2004 and 2024, the journal with the most publications was the *International Journal of Biological Macromolecules* (500 papers), followed by *Food Packaging and Shelf Life* (315 papers) and *Food Hydrocolloids* (245 papers). In addition, according to the Web of Science search, from the 4191 papers, 800 are Reviews and 3312 are Research papers.

Adding the keyword “release” to “active food packaging”, between 2019 and 2024, the Web of Science database returned 1034 results, and the PubMed database returned 649.

The main goal of this review is to compile, between 2019 and 2024, the latest trends in release-type active food packaging, particularly those using EOs, natural extracts, or individual phenolic compounds, to prolong foods’ shelf life. Although this review does not follow a systematic review protocol, a structured and transparent literature search strategy was employed, adhering to PRISMA principles. After an initial search in PubMed and Web of Science using the keywords “active food packaging” and “release,” titles and abstracts were screened to identify studies specifically focused on emission-type active packaging involving essential oils, natural extracts, or individual phenolic compounds. Non-relevant records (e.g., studies on direct food application, gaseous emitters, or innovative packaging without release function) were excluded. Approximately 100+ peer-reviewed articles were selected for detailed analysis.

## 2. Biopolymers

The widespread dependence on conventional petroleum-based plastics has led to significant environmental concerns due to their non-biodegradable nature. As a sustainable alternative, bio-based biodegradable polymers have gained increasing attention. This shift has fueled research into functionalized natural polymers for food packaging, driven by both environmental concerns and the demand for eco-friendly materials. Several natural and semi-synthetic polymers have been investigated for their potential in active packaging systems, including starch [17], pectin [18], polyvinyl alcohol [19], sodium alginate [20], gelatin [21], chitosan [22], polylactic acid [23], carrageenan [24], protein [25], and carboxymethyl cellulose [26].

Among these, PLA stands out as a particularly promising candidate. Synthesized via microbial fermentation of sugars from renewable resources such as sugarcane or corn starch, PLA combines mechanical strength, renewability, transparency, non-toxicity, and processing [27]. Nonetheless, its inherent brittleness and low-impact resistance limit its application in flexible packaging formats [28]. These drawbacks may be mitigated through the incorporation of nanocomposites, essential oils (EOs), natural extracts, or blending with other polymers. However, the incorporation of EOs into PLA can be technically challenging due to potential phase separation, as well as thermal degradation of volatile EO compounds during high-temperature processing such as extrusion or hot pressing.

Chitosan (CS), obtained by the deacetylation of chitin, which is produced by arthropods and fungi, is non-toxic, biofunctional, and biocompatible, and it presents antimicrobial properties [29,30]. Compared to conventional plastics, chitosan offers notable advantages, including biodegradability and derivation from renewable sources. However, it presents some disadvantages, such as inferior mechanical and barrier properties and low solubility in neutral and alkaline conditions, requiring acidic solutions for processing; thus, it is unsuitable for high-temperature applications and has a higher production cost [30]. The acidic medium required for chitosan solubilization may also compromise the stability or dispersibility of certain hydrophobic EOs, affecting film homogeneity and release behavior. The incorporation of EOs, phenolic compounds, and nanoparticles can improve the mechanical performance, reduce the water permeability, and enhance the antimicrobial properties of chitosan-based films [30].

Starch is the most abundant renewable polysaccharide and is widely used due to its film-forming ability, biodegradability, and low cost. Starch-based polymers, such as amylose, which exhibits film-forming capabilities, and amylopectin, which influences crystallinity and mechanical properties, can be processed into films or coatings via casting or extrusion techniques. However, like most natural biodegradable polymers, starch-based materials exhibit several limitations, including low tensile strength, high brittleness, poor barrier properties, and thermal instability at elevated temperatures. Despite their environmental advantages, when compared with conventional plastics, starch-based polymers demonstrate inferior performance characteristics, making them more suitable for short-term packaging applications. Nevertheless, the incorporation of EOs and nanocomposites has been shown to significantly enhance their functional properties and mitigate these drawbacks [31]. From a processing standpoint, the hydrophilic nature of starch and chitosan often leads to poor miscibility with hydrophobic essential oils, which may require emulsifiers or encapsulation strategies to ensure even distribution within the matrix.

When compared directly, PLA offers better mechanical strength and thermal processability, making it suitable for rigid and semi-rigid packaging formats. Chitosan stands out for its intrinsic antimicrobial activity but is limited by its solubility and thermal constraints. Starch is the most affordable and biodegradable but requires significant functional enhancement to match conventional plastic performance. Therefore, polymer selection should be guided by the specific packaging application, shelf-life needs, and processing methods [32].

EOs offers a natural approach to controlling microbial growth and preserving food quality. Incorporating EOs into films and coatings offers a practical advantage over direct application. This method allows for controlled release of bioactive compounds, prolonging their contact with the food and mitigating the potential negative impact on sensory attributes that can occur with direct application of higher, effective doses that may surpass acceptable limits.

Despite the promise of bio-based materials, many remain limited to laboratory-scale applications due to challenges in thermal processability and mechanical stability [33]. Consequently, natural polymer composite films functionalized with EOs have become a major focus of research in active packaging. A wide range of EOs have been explored for this purpose, including cinnamon, clove, tea tree, basil, eucalyptus, thyme, rosemary, sage, *Artemisia absinthium*, pimento, coriander, lime, fennel, oregano, lavender, turmeric, and orange, among others [17,26,34,35,36,37,38,39,40,41].

Effective active packaging requires a gradual and sustained EO release to maintain antimicrobial and antioxidant activity throughout the product’s shelf life. Nanoparticles have shown promise in controlling EO release by acting as a physical barrier to additive release [36,42]. Surendhiran et al. [36] reported that the tortuous release pathway of EOs within a nanomatrix into bio-nanocomposite film (encapsulated essential oil) results in slow release, thereby extending food shelf life. In contrast, simply blending EOs into packaging films leads to a rapid and uncontrolled release of the EO over a short period, reducing the preservation time.

The growing interest in biopolymer-based systems functionalized with natural compounds has led to a proliferation of studies testing these materials in real food matrices, which are discussed in Section 4.

## 3. Active Compounds

### 3.1. Essential Oils

#### 3.1.1. Sources and Composition

EOs are volatile, aromatic, oily liquids extracted from various plant parts [43]. A single plant species may produce multiple types of EOs, located in different parts such as the leaves, flowers, buds, shoots, peels, bark, seeds, rind, and roots [20,34,44,45,46].

EOs are rich in bioactive compounds, exhibiting both antioxidant and antimicrobial activities. They represent a generally recognized safe and biodegradable alternative to synthetic additives, as supported by their GRAS (Generally Recognized as Safe) status, and demonstrated efficacy in several systematic reviews and meta-analyses [5,32,43,47]. These natural products possess a complex composition, with terpenes being the most common class of chemical compounds present [44].

#### 3.1.2. Extraction Methods

The conventional methods for producing essential oils include distillation (hydrodistillation and steam distillation) and solvent extraction. Cold pressing is typically used for citrus essential oils [48]. More advanced techniques, such as supercritical fluid extraction, microwave-assisted extraction, and ultrasound-assisted hydrodistillation (sonohydrodistillation) have also been utilized [34,49,50].

A summary of the most widely applied extraction methods for obtaining essential oils is presented in Figure 1.

#### 3.1.3. Bioactivity and Mechanisms of Action

EOs exhibit a wide range of biological activities, including antimicrobial, antioxidative, and anti-deterioration effects, contributing to extended food product shelf life [19,20,21,23,44]. These oils achieve this by controlling microbial growth and preserving food integrity. While the principal components of EOs are generally considered the primary contributors to their bioactivity, minor constituents may also exert significant influence through synergistic or additive interactions [51]. The potential synergistic effects of the combination of plant essential oil (EO) offer a promising strategy to enhance their action.

Studies have shown correlations between EO composition and bioactivity. Chen et al. [6] found a strong positive correlation between antioxidant activity and eugenol and thymol content in cinnamon, thyme, and clove oils. In contrast, lavender and peppermint oils, rich in linalool and menthol, showed low antioxidant activity. Similarly, Manssouri et al. [52] observed a positive correlation between antioxidant activity and phenolic compound content in *Ammodaucus leucotrichus* Coss. & Dur. fruit EO. Lee et al. [53] attributed the high antioxidant activity of *Citrus unshiu* (linalool-rich), and *Cinnamomum loureirii* Nees (eucalyptol, α-citral, β-citral, and linalool-rich) oils to their compositions. This compositional difference also correlated with antimicrobial activity; *C. unshiu* oil showed no such activity, whereas *C. loureirii* oil exhibited strong antifungal activity against *Candida albicans*.

Several studies have investigated the antimicrobial activity of EOs against various foodborne pathogens. Yasir et al. [54] investigated the antibacterial activity of several essential oils against pathogens isolated from raw milk. Thyme oil (*Thymus vulgaris*) exhibited the strongest antimicrobial activity overall, particularly against *Escherichia* spp. Oregano oil also showed strong activity against *Escherichia* spp., suggesting a possible synergistic effect when used in combination with thyme oil. Lemongrass oil was highly effective against *Staphylococcus* spp. Rosemary (*Rosmarinus officinalis*) and Spearmint mint (*Mentha spicata*) oils showed the lowest antibacterial activity. The antimicrobial action of species was attributed to the presence of different active compounds, mainly phenolic compounds.

Mangalagiria et al. [55] reported Lemongrass and palmarosa oils demonstrated stronger antibacterial activity than other oils (eucalyptus, geranium, citronella, and tagetes), with citral (a major component of lemongrass oil) exhibiting potent bactericidal activity. They also found these oils to be effective against four fungi tested, *Aspergillus niger*, *Fusarium oxysporum*, *Fusarium udum*, and *Magnaporthe grisea*. Abers et al. [45] screened 19 volatile compounds from various EOs and found rosemary, tea tree, and cassia oils to be the most potent broad-spectrum antibacterial agents. In contrast, clove volatiles exhibited virtually no antimicrobial activity.

The antimicrobial and antioxidant effects of essential oils are primarily attributed to specific classes of compounds, including phenolics, terpenes, and aldehydes, each acting through distinct yet sometimes overlapping mechanisms. Phenolic compounds such as thymol, carvacrol, and eugenol integrate into bacterial membranes, disrupting their lipid bilayers and leading to increased permeability, ion leakage, and eventual cell lysis. These compounds can also scavenge free radicals and chelate metal ions, thus preventing oxidative damage. Terpenes (e.g., limonene, pinene) mainly increase membrane fluidity and may interfere with intracellular enzymes. Aldehydes like cinnamaldehyde alter membrane protein conformation and inhibit enzyme activity by reacting with thiol groups. Figure 2 summarizes these major mechanistic pathways.

The antimicrobial activity of EOs involves multiple mechanisms, including disruption of microbial cell membranes, leakage of intracellular contents, generation of reactive oxygen species (ROS), and inhibition of key metabolic enzymes. Phenolic compounds such as carvacrol and thymol integrate into lipid bilayers, increasing membrane permeability, while eugenol can interfere with ATP production. These mechanisms contribute to broad-spectrum bioactivity against spoilage and pathogenic organisms.

Microbial inactivation by EOs is a complex process influenced by several factors such as the microorganism, EO properties (type and concentration), the food matrix, and the treatment intensity [56]. Furthermore, the chemical composition and content of EO are subject to variation due to both intrinsic (genetic characteristics and phenological stage) and extrinsic factors (environmental conditions, geographical location, seasonality, soil and the method of extraction) [50,57,58,59,60,61,62].

Although phenolics are major compounds of EOs, there are others with equally powerful biological activity, such as terpenes and terpenoids, aldehydes, ketones, alcohols, esters, and ethers [63,64].

#### 3.1.4. Applications for Active Packaging

##### Innovations in Encapsulation and Controlled Release

Incorporating EOs into food packaging materials is a significant strategy for extending the shelf life of perishable foods while addressing consumer concerns about the potential health risks associated with synthetic additives [19,20,21,23].

In the development of a new generation of food packaging, the focus is on environmentally friendly materials from renewable sources. Flexible bio-based films, primarily composed of biopolymers like polysaccharides, proteins, and lipids, provide a sustainable matrix for natural active agents. Essential oils offer a natural solution for microbial control and food preservation in bio-based films.

Nanoencapsulation, emulsification, and cyclodextrin inclusion are widely explored for enhancing EO stability, reducing volatility, and enabling controlled release.

##### Synergistic Combinations and Functional Blends

EOs can also contribute to desirable sensory qualities, such as color and flavor, enhancing freshness [65]. For example, *Zanthoxylum bungeanum* EO improved color properties and inhibited lipid oxidation in Chinese-style sausage [66]. Lima et al. [25] developed an aromatic and antioxidant active film based on myofibrillar protein from fish by-products and passion fruit EO, where the EO provided antioxidant activity and a desirable aroma while reducing fish odor. A review by Zhang et al. [65] highlighted the positive influence of EOs on food sensory characteristics.

Several studies also explore synergistic blends, such as thyme–oregano or clove–cinnamon oil combinations, which enhance antimicrobial efficacy while reducing the amount of EO needed.

##### Polymer–EO Interactions and Performance

Flexible biopolymer matrices can influence the activity and stability of incorporated EOs, but the interaction between EOs and polymers also affects film structure and function.

EO incorporation may alter polymer properties, such as increasing film brittleness or lowering tensile strength. Formulation strategies (e.g., blending, use of plasticizers) are often needed to balance functionality and mechanical integrity.

In summary, recent studies between 2019 and 2024 highlight key innovations in emission-type active packaging with essential oils. Encapsulation techniques, synergistic blends, and polymer–EO optimization are among the most studied strategies. While the effectiveness of EOs against spoilage organisms and oxidation is well documented, performance trade-offs—such as mechanical weakening or aroma intensity—are common. Furthermore, many studies are limited to in vitro systems, with few addressing food matrix effects, sensory impact, or long-term migration, pointing to the need for more application-oriented research.

#### 3.1.5. Limitations and Technological Solutions

Nevertheless, the use of EOs in food preservation is sometimes limited by their intense aroma. The concentrations required for effective antimicrobial or antioxidant activity can negatively impact the food’s organoleptic properties [67].

However, free EOs are limited by low water solubility, oxidation, and volatilization. Nanoencapsulation has proven to be a valuable strategy for enhancing the stability and bioactivity of EOs compared to their free form [68]. Incorporating EOs into bio-based films via Pickering emulsions also offers effective solutions, enhancing antimicrobial and antioxidant activities and leading to extended food shelf life [69].

Beyond volatility and aroma issues, critical challenges include ensuring compatibility between EOs and food matrices, controlling migration rates within legal limits, and addressing regulatory barriers—particularly for compounds not classified as GRAS or exceeding specific migration limits (SMLs). Moreover, consumer perception remains a key barrier, as strong EO aromas may affect sensory acceptance unless properly masked or stabilized.

#### 3.1.6. Patent Developments

Numerous patents cover the use of essential oils in food packaging, reflecting the positive results achieved with this approach (Table 2).

One patent describes a protein-based edible film designed for efficient essential oil release (U.S. Patent 20210186064). Another invention details a process for creating essential oil-modified nanocellulose, where the essential oil is covalently bonded to the nanocellulose to prevent leaching, resulting in edible coatings with prolonged antimicrobial effects (U.S. Patent 20210054104). A Chinese patent (CH116478546A) describes a controlled-release antibacterial active packaging film. This film uses composite nanoparticles (zein and carboxylated cellulose nanowhiskers) to coat essential oil droplets, forming a stable dispersion. This emulsion is then incorporated into the film, allowing for controlled release of the essential oil. Another Chinese patent (CN 114249926B) discloses edible packaging films based on corn starch and chitosan incorporated with essential oil microcapsules.

While these technological advancements and patent developments reflect significant progress in the field, broader implementation still faces several non-technical challenges related to regulation, economics, and consumer acceptance, as discussed below.

#### 3.1.7. Regulatory, Economic, and Consumer Acceptance Challenges

Several barriers hinder the transition from laboratory research to commercial application of EO-based active packaging. Regulatory approvals, such as GRAS status in the United States and European Food Safety Authority (EFSA) evaluations in the European Union, are essential for ensuring food contact compliance. These assessments require detailed toxicological, chemical, and migration data, which are often lacking or difficult to obtain for complex mixtures, such as EOs [70,71]. Furthermore, the regulatory landscape is not harmonized globally, creating additional complexity for multinational commercialization [71].

Economic constraints, particularly the cost of high-purity EOs and encapsulation technologies, can limit scalability. While encapsulation enhances the stability and controlled release of volatile compounds, it adds to production costs and may require specialized equipment [70]. Small and medium enterprises (SMEs) may find it challenging to invest in such technologies without clear market incentives or regulatory facilitation [71].

Consumer acceptance is another critical factor. Sensory attributes such as odor intensity and product appearance influence purchase decisions. Some consumers may perceive EO-based packaging as unnatural or intrusive, especially if the labeling is unclear or implies the presence of functional additives in the food. Transparent communication and education about the benefits and safety of such packaging systems are essential for building trust. Moreover, studies show that while consumers often express interest in sustainable and natural packaging, actual willingness to pay more remains limited [72].

Addressing these challenges requires a multi-stakeholder approach, involving collaboration among industry, regulatory bodies, and consumer advocacy groups. Innovations in cost-effective encapsulation methods, harmonization of safety assessments, and strategies for consumer engagement will be crucial for the successful implementation of emission-type active packaging solutions in the food sector [71,72].

### 3.2. Natural Extracts

#### 3.2.1. Sources and Composition

Besides EOs, natural extracts can be obtained from plants, animals, herbs, spices, or microflora, as well as agro-industrial by-products [73,74].

These natural extracts are rich in different compounds, such as flavonoids, phenols, organic acids, indicans, vitamins, terpenoids, protein, amino acids, sugar, glycoproteins, resins, lignins, fats, alkaloids, and colloids [75,76].

#### 3.2.2. Extraction Methods

Another factor that can affect the composition of the natural extract is the extraction process. There are different extraction techniques with different impacts on the end-product and the environment. The traditional extraction techniques include Soxhlet extraction, solvent extraction, and steam distillation, which are environmentally harmful, expensive, and time-consuming [77]. The greener extraction techniques include microwave-assisted extraction (MAE), ultrasound-assisted extraction (UAE), supercritical fluid extraction (SFE), and pressurized liquid extraction (PLE) (Figure 3).

Compared to traditional extraction techniques such as maceration or hydrodistillation, green methods like microwave-assisted extraction (MAE) and ultrasound-assisted extraction (UAE) offer clear environmental advantages. Life cycle assessment (LCA) studies have demonstrated that both MAE and UAE significantly reduce energy consumption, greenhouse gas emissions, and water use per unit of bioactive compound extracted, with MAE ranking among the most eco-efficient methods [78]. Additional comparative analyses show that these techniques can lower processing time and solvent usage by up to 40–60% while maintaining or even improving extraction yields compared to conventional methods [79]. Such evidence helps to contextualize and validate the sustainability claims often associated with green extraction, underscoring the relevance of MAE and UAE as truly eco-friendly solutions for recovering bioactives in active packaging applications.

These techniques are fast, environmentally friendly, and do not use harmful solvents, producing extracts with higher quality and yield [77]. The use of natural extracts in the development of food packaging comes from a circular economic perspective, since natural extracts can be derived from food waste, often rich in bioactive compounds, leading to zero waste.

#### 3.2.3. Bioactivity and Mechanisms of Action

Natural extracts can have different properties, like antioxidant, antimicrobial, antifungal, and anti-browning activity, in addition to other biological effects such as anti-hypertensive, anti-inflammatory, anti-anxiety, anti-cancer, and anti-diabetic effects [74,77,80]. Nonetheless, these properties are dependent on the composition of the natural extracts, which can vary on several factors, including variety, geographical origin, development stage, climate conditions, the season of growth, ripening stage, and harvest [81].

#### 3.2.4. Applications for Active Packaging

Over the past five years, numerous studies have explored new active food packaging systems incorporating natural extracts (Section 4). Several bio-degradable polymers have been studied for this purpose, including chitosan, alginate, poly (lactic acid) (PLA), carrageenan, cellulose, whey protein, soy protein, and carboxymethyl cellulose, among others. The combination of these polymers with active natural extracts offers multiple benefits. For instance, using natural extracts can reduce the need for synthetic additives directly added to foods, contributing to a healthier diet. Furthermore, adding these natural extracts as active components to the food packaging has been demonstrated to improve the packaging properties.

For instance, Mellinas et al. [82] developed and characterized a pectin-based active film with zinc oxide nanoparticles and cocoa bean shell extract obtained from food by-products. The addition of the natural extract and zinc oxide nanoparticles improved the overall oxygen, thermal, and UV barrier properties of the films, where the oxygen barrier was improved by 50% and the screen to UV radiation reached 98% [68]. In another study by Rambabu et al. [83], chitosan films incorporated with mango leaf extract were developed and analyzed. In this study, the addition of mango leaf extract increased the film’s thickness, density, and surface hydrophobicity. It also improved the film’s antioxidant activity and tensile strength, but it decreased the film’s water vapor permeability, water solubility, and elongation at break [83]. Todhanakasem et al. [84] studied the film composed of polyvinyl alcohol, corn starch, glycerol, and watermelon rind extract as the active compound. The active film presented higher tensile strength and elongation at break and improved water vapor permeability compared to the control film (without extract). The active film also demonstrated excellent antioxidant capacity and antimicrobial properties [84].

In addition to evaluating the effect of natural extracts on the food packaging developed, studies have been developed to determine the impact of this active packaging on food. For example, Andrade et al. [85] evaluated the effectiveness of a whey protein film incorporated with rosemary extract against the lipid oxidation of salami during 90 days at 5 °C. The authors concluded that the active film was able to delay lipid oxidation of salami, as the samples presented lower values of malonaldehyde (MDA) and hexanal in comparison with the control. All samples presented levels of MDA below 0.5 mg MDA/kg, the value at which consumers do not detect the off flavor [85]. In another study, the effectiveness of a chitosan–starch coating with garlic extract incorporated on green and yellow bell peppers (*Capsicum annuum*) over 18 days was evaluated [86]. The coating effectively prevented weight loss and bacterial growth during the storage period, demonstrating its potential in maintaining both green and yellow bell pepper properties [86]. A chitosan-peony (*Paeonia suffruticosa* Andr.) leaf extract composite film was developed by Ma et al. [87] and applied to apples to evaluate its potential in extending fruit shelf life. In addition to the positive effects of the extract on the physical and barrier properties of the film (Section 4), the active film was able to protect the fruit from natural browning, compared to the control, as the film protected the fruit from oxidation and external factors [87]. Fan et al. [88] successfully developed an active film packaging based on chitosan and starch and with *Portulaca oleracea* extract incorporated. The *Portulaca oleracea* chitosan–starch film successfully protected the meat colors and overall appearance during storage. The active film was able to delay the meat from lipid oxidation and spoilage compared to the meat packed with the control film and polyethylene film [88].

In summary, polysaccharide-based films—particularly those based on chitosan and starch—demonstrated high compatibility with phenolic-rich plant extracts, showing consistent antimicrobial and antioxidant activity, especially against fungal spoilage and lipid oxidation. Extracts from grape seed, pomegranate peel, and oregano emerged as the most effective, often reducing microbial counts by 2–3 log CFU/g or delaying oxidation in high-fat foods. However, trade-offs were frequently observed; films with high bioactivity often showed reduced mechanical strength or increased brittleness, especially in systems lacking adequate plasticizers or encapsulation strategies. Additionally, several studies relied on in vitro tests, with limited validation in real food systems or sensory analyses, underscoring the need for more holistic assessments in future research.

Further applications, including meat, dairy, and fresh produce preservation using grape seed extract-, oregano oil-, and chitosan-based films, are detailed in Section 4.

### 3.3. Phenolic Compounds

The biological activity of natural extracts and EOs is largely attributed to their phenolic content. Phenolic compounds constitute a broad class of secondary metabolites, encompassing multiple sub-classes with distinct structural and functional properties. Despite this diversity, only a limited number of phenolics have been extensively studied for applications in active food packaging (Table 3).

#### 3.3.1. Bioactivity and Mechanisms of Action

The antioxidant activity of phenolic compounds is primarily exerted through their interaction with free radicals via several mechanisms: (1) hydrogen atom transfer (HAT)—the donation of hydrogen atoms from phenolic hydroxyl (-OH) groups to neutralize free radicals; (2) single-electron transfer (SET)—reduction in radical species (e.g., Fe^3+^, Cu^2+^) via electron donation; (3) sequential proton loss electron transfer (SPLET)—deprotonation of the -OH group followed by electron transfer, forming resonance-stabilized phenoxyl radicals, or (4) transition metal chelation—complexation of redox-active (Fe^2+^, Cu^+^) [89].

In terms of antimicrobial activity, phenolic compounds have several antimicrobial mechanisms, all centered on damaging the membrane cell wall, allowing for the entrance of the compound in the cell, disturbing intracellular functions, and inducing apoptosis [7]. Their mode of action can vary significantly depending on their structure, which has prompted research into synergistic combinations of phenolics to enhance bioactivity.

Cinnamaldehyde is one of the main compounds of cinnamon, specially found in the varieties *Cinnamomum verum* and *Cinnamomum cassia*. Responsible for the sweet taste of cinnamon, this hydrophobic aromatic aldehyde is obtained from the bark (main compound), leaves, and roots of trees of the genus Cinnamomum [90]. It exhibits significant anti-cancer, anti-inflammatory, antioxidant, and anti-angiogenic activities, modulating cellular processes such as apoptosis, oxidative stress, and immune responses. Its main applications include potential use in cancer therapy, food flavoring, and as a therapeutic agent for diabetes and cardiovascular diseases [91,92,93].

#### 3.3.2. Applications for Active Packaging

In the context of food packaging, Aragón-Gutiérrez et al. [94], incorporated cinnamaldehyde into ethylene vinyl alcohol-based films via solvent casting and melt extrusion at concentrations of 5, 10, and 15%. The resulting active films demonstrated antioxidant activity by the DPPH radical scavenging assay and antimicrobial activity against *Penicillium expansum* [94]. Similarly, Yu et al. [95] incorporated cinnamaldehyde, carvacrol, and eugenol into zein-based films by solvent casting at 1, 3, and 5%. The film with 5% of carvacrol showed antimicrobial activity against *S. aureus* and *E. coli* [95]. Further applications of cinnamaldehyde and related phenolic compounds in active packaging are summarized in Section 4.

#### 3.3.3. Limitations and Technological Solutions

Despite their functional potential, phenolic compounds face challenges in packaging applications. Sensory impact is important in the acceptance of the final products by the consumer, and several phenolic compounds have strong aromas or colors that could impact the final product. Additionally, their high volatility limits their effectiveness in active food packaging as rapid, rather than controlled, migration fails to extend shelf life adequately.

To overcome these limitations, encapsulation techniques using cyclodextrins or liposomes offers a solution by reducing migration rates. Additionally, polymer selection plays a key role; using neutral-flavor matrices like polylactic acid or chitosan can mitigate sensory interference. Finally, optimizing concentrations ensures maximal functional benefits while maintaining sensory acceptability.

## 4. Examples of Application of New Active Food-Packaging Materials

Active food packaging incorporating EOs, natural extracts, or phenolic compounds can be tested on perishable foods like cheese, fruits, fish, and meat. These systems have consistently demonstrated the ability to inhibit microbial growth and lipid oxidation, contributing to shelf-life extension and improved food quality [17,39,96,97].

In one study, pork patties were packaged using CS films incorporating clove oil (CO) and/or nisin (NI). The CS-CO-NI film best preserves its appearance and slows down redness changes, likely due to the antioxidant and antimicrobial activities of CS, CO, and NI. CO proved most effective in inhibiting metmyoglobin formation and lipid oxidation, as indicated by free fatty acid, peroxide value, and thiobarbituric acid reactive substances. Microbiologically, pork patties packed with CS-CO-NI showed the strongest inhibition of pseudomonas, enterobacteriaceae, and lactic acid bacteria, suggesting synergistic antimicrobial effects of the combination of CS, CO, and NI [98].

Similarly, CS films incorporating turmeric essential oil (TEO) were evaluated for surimi preservation [36]. While a CS/TEO film initially inhibited Bacillus cereus growth, its effect diminished after 6 days due to the uncontrolled, speedy release of TEO. A CS/ magnetic nanoparticles/silica/TEO bio-nanocomposite film (chitosan-based food packaging incorporating encapsulated TEO into magnetic nanoparticles/silica nanocomposites), provided sustained inhibition, maintaining low bacterial counts throughout the 14-day storage period due to controlled TEO release. Both films helped maintain a near-neutral pH, suggesting that TEO has antioxidant properties. Minimal silica migration (0.70%) was observed from the bio-nanocomposite film into the surimi, with no detectable MNPs [36].

In the case of fruits, strawberries coated with a yam starch base incorporating lime, fennel, and lavender essential oils were stored at 25 °C and 85% relative humidity for two weeks. These coated strawberries exhibited less weight loss and reduced *Aspergillus niger* growth until day 8 of storage. Among the tested oils, lavender EO demonstrated the highest antioxidant activity [17].

Cheese products have also benefited from active coating. Double cream cheese coated with a film composed of achira starch and microcrystalline cellulose, containing either garlic oil (EXg) or oregano oil (EXo), was compared to uncoated cheese (C1) and cheese coated without any natural oils (C2). EXo samples exhibited the lowest weight loss and the least hardness throughout the storage period. Both EXg and EXo coatings minimized color changes and effectively maintained the product’s microbiological quality; furthermore, they improved the sensory acceptance of the cheese. The oregano oil coating proved most effective overall [99].

Taken together, the results from Table 4, Table 5 and Table 6 suggest that the combination of chitosan with clove oil or turmeric EO, and PLA with oregano or cinnamon oil, were among the most effective in extending shelf life, reducing microbial loads, and maintaining sensory attributes. For instance, the use of CS–clove oil–NI films extended the shelf life of pork by up to 12 days [98], while CS-based bionanocomposites incorporating encapsulated turmeric EO sustained microbial inhibition for 14 days in surimi [36]. PLA films with thymol- or oregano-encapsulated systems preserved the freshness of fruit and meat for up to 30 days [100,101]. In terms of natural extracts, rosemary, green tea, and garlic extracts demonstrated strong antioxidant and antimicrobial capacities, prolonging the stability of products in meat and bakery items for up to 30 days [85,86,102,103,104]. Films that allowed for gradual and controlled release—such as those incorporating encapsulated EOs or nanoemulsions—consistently outperformed direct incorporation methods. Sensory evaluations, when performed, generally indicated improved consumer acceptance with EO-based films compared to controls.

These findings underscore the importance of selecting suitable combinations of biopolymer matrix and active agents, as well as the use of encapsulation strategies to ensure sustained protection throughout the product’s shelf life. Altogether, the results highlight the potential of biopolymer-based films functionalized with natural compounds in extending shelf life and maintaining the quality of various perishable food products. Additional examples of active packaging applications using essential oils, natural extracts, and individual compounds are summarized in Table 4, Table 5 and Table 6, respectively, and support the trends discussed above.

## 5. Conclusions

Active food packaging technologies, particularly those incorporating emission-active compounds such as essential oils (EOs), natural extracts, and individual phenolic compounds, have emerged as innovative tools in food preservation over the last five years. These compounds exhibit strong antimicrobial and antioxidant functionalities that can be effectively deployed via controlled-release packaging systems. When embedded in biodegradable and bio-based polymers like chitosan (CS), polylactic acid (PLA), or starch, they not only enhance food safety and shelf-life but also reduce dependency on synthetic additives and non-degradable plastics.

This review demonstrates that innovations in encapsulation methods, such as nano-encapsulation, emulsions, and cyclodextrin complexes, are critical in mitigating limitations like volatility, rapid degradation, and sensory interference. These methods enable a gradual release of active compounds, essential for maintaining the functional efficacy of the packaging throughout storage and distribution.

Furthermore, the field has seen significant strides in using agricultural and food processing by-products to obtain natural extracts. This approach not only reduces waste but also extends the value chain of food production systems by converting residues into functional packaging components. This aligns with circular economy principles, which emphasize resource efficiency, waste minimization, and material circularity, and contributes to the valorization of food waste, a critical goal in global sustainability efforts. The integration of these bioactives into biodegradable films contributes to more sustainable packaging solutions by reducing reliance on virgin raw materials and fossil-based plastics. Several studies have successfully integrated these bioactive materials into films that simultaneously improve the mechanical, barrier, and optical properties of packaging while offering potent biological effects against spoilage organisms and oxidative processes.

The commercial potential of these materials is underscored by the growing number of patents and pilot-scale developments observed during the review period. However, a clear gap remains between laboratory innovation and industrial implementation. Scaling up these technologies requires addressing regulatory approval, consumer acceptance, cost-effectiveness, and consistent raw material sourcing.

Additionally, long-term toxicological studies and comprehensive environmental impact assessments of both the active compounds and the packaging materials themselves are essential for broader market acceptance. Regulatory frameworks must evolve to keep pace with these innovations, providing clear guidance for using natural and biodegradable materials in food-contact applications.

The effectiveness of active packaging systems is also strongly influenced by the food matrix itself. Characteristics such as fat content, moisture level, and pH can affect the release kinetics, stability, and bioactivity of the compounds. For instance, high fat content may enhance the migration of lipophilic molecules, while acidic environments can promote or degrade specific phenolic structures. These interactions must be considered when designing tailored packaging solutions for specific food types. Looking ahead, efforts should be directed toward the development of open source databases of validated formulations, the standardization of migration and sensory testing methods, and the creation of regulatory roadmaps to facilitate market entry of bioactive films. Intelligent packaging integrating active functions with sensorial or indicator technologies may also offer valuable synergies in quality control. In the future, interdisciplinary collaboration between food scientists, materials engineers, chemists, and regulatory bodies will be essential to bridge the gap between academic research and commercial viability. However, these collaborations should be focused on specific goals such as scaling encapsulation methods, optimizing formulations for different food matrices, and addressing safety and legislative barriers.

Ultimately, emission-active packaging represents a promising pathway toward safer, more sustainable food systems. To fully realize this potential, concrete actions must be taken at both the scientific and regulatory levels, aimed at improving functional performance, safety, and consumer acceptance. These objectives are inherently aligned with the circular economy framework, which promotes material reuse, biodegradability, and the functional recovery of resources within the food-packaging–waste cycle.

Despite the promising advances reported in recent years, it is important to highlight certain methodological limitations observed in the reviewed studies. A significant number of works are based on in vitro conditions, which may not fully replicate the complexity of real food systems or commercial storage environments, limiting their applicability at the industrial scale. Sensory evaluations involving consumer panels are often absent, making it difficult to predict acceptance of the final product. Furthermore, the lack of standardized methodologies for extraction, encapsulation, film formation, and release kinetics hampers direct comparison between studies. Toxicological assessments related to the long-term effects of active compound migration are also scarce, as are comprehensive studies on environmental impact. Addressing these limitations through interdisciplinary collaboration, real-world validation, and data sharing will be crucial to bring these packaging systems closer to industrial adoption.

## Figures and Tables

**Figure 1 foods-14-02713-f001:**
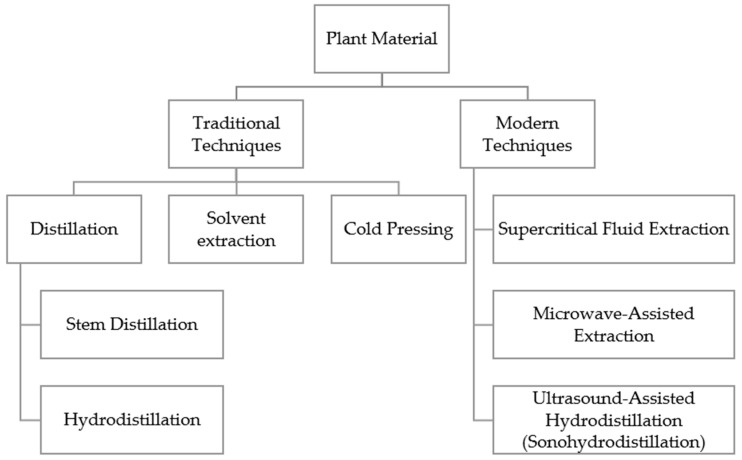
Principal extraction methods for obtaining essential oils from plants.

**Figure 2 foods-14-02713-f002:**
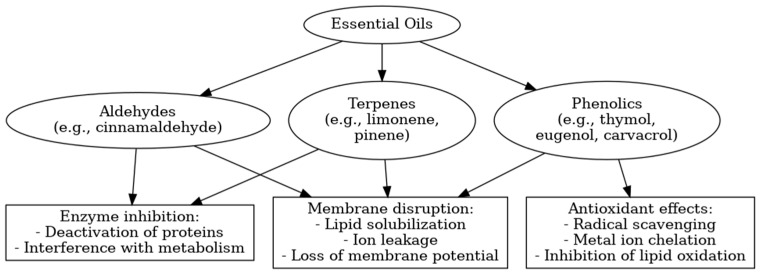
Schematic representation of the mechanisms of action of essential oil constituents in food preservation.

**Figure 3 foods-14-02713-f003:**
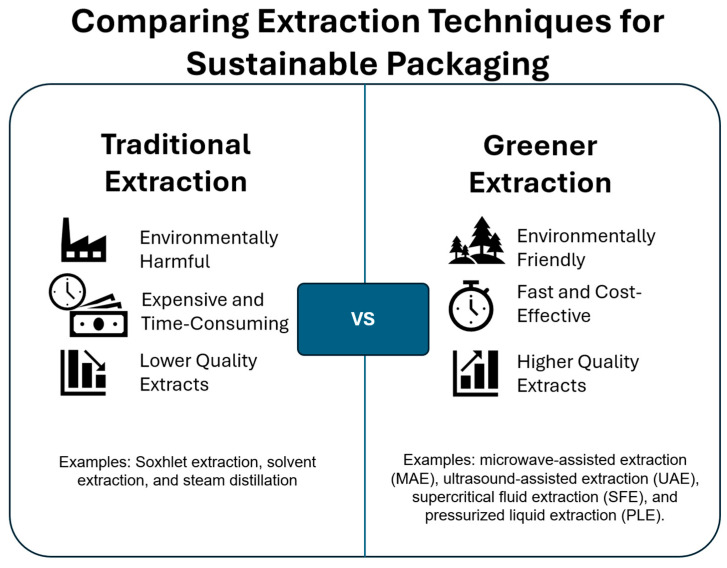
Comparison between traditional and greener extraction methods for natural extracts. Greener methods such as MAE and UAE offer advantages, including reduced solvent use and shorter processing time.

**Table 1 foods-14-02713-t001:** Summary of Publication Trends on Active Food Packaging (2004–2024).

Database	Search Keywords	Total Results	Results (2019–2024)	Results (2023–2024)	Top Journals	Article Types
PubMed	active food packaging	6320	4428	2020	*Int. J. Biol. Macromol.* (1045), *Food Chem.* (379), *Polymers* (347)	Not specified
Web of Science	active food packaging	5740	4191	1968	*Int. J. Biol. Macromol.* (500), *Food Packag. Sheld Life* (315), *Food Hydrocolloids* (245)	Reviews: 800 Research papers: 3312

**Table 2 foods-14-02713-t002:** Patents covering the use of essential oils in food packaging.

Patents	Title	Year of Publication
CN110396224A	A preparation method of anti-oxidative and antibacterial film equipped with cinnamon essential oil Pickering emulsion	2019
CN108752610B	A kind of edible antibacterial film of slow-release essential oil and preparation method thereof	2020
AU2020103921A4	Antibacterial compound fiber film, method for preparing the same and application thereof	2020
CN113354853B	A kind of biodegradable high barrier antibacterial composite film and preparation method thereof	2022
CN114411331A	Nanometer film added with oregano essential oil cyclodextrin inclusion compound and preparation method and application thereof	2022
CN113846490B	Preservative film with intelligent microbial self-inhibition effect and preparation method thereof	2022
CN111440344B	A kind of composite seaweed polysaccharide-based double-layer packaging film and its preparation method and application	2022
CN113956633B	A degradable active packaging material based on ginger essential oil and polylactic acid and its preparation method thereof	2022
CN113712072B	A clove essential oil Pickering emulsion film preservative and its preparation method and application	2023
CN114249926B	Edible film and preparation method thereof	2023
JP2024513989A	Edible spray or coating compositions and methods of making and uses thereof for extending the shelf life of perishable items	2024
CN115516046B	Coating for food packaging comprising antimicrobial active ingredient	2024
KR102735729B1	Paper wrapper for packing food with excellent fragrance activity and antimicrobial activity	2024
CN113907128B	Edible pepper essential oil film for fruit and vegetable fresh-keeping and preparation method thereof	2024

**Table 3 foods-14-02713-t003:** Number of search results combining each individual compound with the keywords ‘active food packaging’ (2019–2024).

Compound	PubMed	Web of Science	Scopus
Thymol	87	125	466
Cinnamaldehyde	68	112	420
Limonene	38	37	257
Carvacrol	80	161	483
Eugenol	46	83	376
Vanillin	19	15	145
Linalool	29	29	180
Citral	18	31	138
α-Pinene	7	8	117
Citric acid	65	84	567
Gallic acid	91	100	582
Quercetin	50	56	397
Resveratrol	16	14	172
Catechin	39	37	305
Epicatechin	42	12	122

**Table 4 foods-14-02713-t004:** Studies with active packaging examples using essential oils as active components.

Essential Oil	Polymer	Packaging Preparation Method	Food Matrix (Applied)	Main Results	Ref.
*Mentha piperita*; *Mentha* x *villosa* Huds	Chitosan (CS)	Coatings	Papaya (*Carica papaya* L.)	EOs were incorporated into chitosan-based coatings to inhibit fungi growth during papaya storage in refrigerators. Formulated coatings did not affect papaya sensory acceptability.	[105]
Basil, coriander, pimento, rosemary, thyme	Whey protein isolate	Coating	Sliced bologna-type sausage	Coatings incorporating pimento EO provided the most effective inactivation against *Listeria innocua* followed by thyme, basil, coriander, and rosemary EO. Coatings containing thyme essential oil were the best sensory coating types.	[35]
Clove (C)	CS	Casting	Pork patties in cold storage	Based on sensory and microbiological evaluations, the shelf life of pork patties was 6 days for control, 9 days for CS and CS-Nisin, and 12 days for CS-C and CS-C-Nisin (CS-C-NI). C showed high antioxidant activity, and the combination of C and CS may enhance oxidative stability of pork patties during storage. CS-C-NI combination treatment has excellent microbial inhibition due to synergistic bactericidal effects.	[98]
Thyme (encapsulated into porous poly (lactic acid) nanofibers—PLA)	Poly(vinyl alcohol)/poly(ethylene glycol)—(PVA/PEG)	Electrospinning	Strawberries	Thyme essential oil significantly inhibited bacterial survival in vitro. The slower release of TEO from the PLA/TEO/PVA/PEG composite films, compared to the PLA/TEO nanofibers, contributed to the extended shelf life of the strawberries. PLA/TEO/PVA/PEG film shows higher microbial activity against Escherichia coli and *Staphylococcus aureus*.	[100]
Oregano (free and microencapsulate)	Wheat flour and poly (butylene co-terephthalate adipate)	Blown extrusion	Brazilian fresh pastry (known as pastel)	Fresh pastries packaged with film incorporating oregano essential oil microparticles exhibited lower mold and yeast counts during 28 days of refrigerated storage compared to those packaged with control film or film containing free oregano EO. This may have occurred due to the slow and gradual migration of the OEO from the film to the food surface.	[101]
Turmeric (TEO); turmeric (encapsulated into magnetic/silica porous core–shell nanocomposites-MNPs/Si)	CS	Casting	Surimi	The CS/TEO film effectively inhibited Bacillus cereus growth, significantly reducing the bacterial population for up to 6 days of storage. However, due to the rapid and uncontrolled release of TEO when directly incorporated into the film, bacterial growth resumed. In contrast, the CS/MNPs/Si/TEO film maintained reduced bacterial proliferation, until the end of storage (14 days), likely due to the slow release of TEO. Both films prevented surimi protein oxidation, suggesting TEO’s antioxidant potential.	[36]
Clove	Cellulosic nanocrystals obtained from the Kudzu plant (*Pueraria montana*), and corn starch	Casting	Red grapes	The films loaded with essential oil exhibited remarkable antimicrobial properties against *S. aureus* and *E. coli*. The antimicrobial effect was stronger on *S. aureus*. In addition, films with essential oils were found to be more efficient in maintaining the fruit’s physical and chemical stability for 15 days at 5 °C.	[106]
Clove (doubly stabilized oil chitosome nanoparticles (CNPs))	Gelatin/PVA (GEL/PVA)	Casting	Marinated steaks	The presence of CNPs in film suppressed microbial proliferation, decelerated meat product degradation, and preserved the color and freshness of the meat products during storage. This was attributed to the antimicrobial effect of the CNPs.	[107]
*Trachyspermum ammi*	PLA	Tape casting	Waffles	Waffles packed in PLA films containing 50 wt% blend of both oils had their shelf life extended up to 30 days compared to 2 days for the neat PLA film. Vanilla was found to be effective in masking the unpleasant odor of Tammi.	[23]
*Cinnamon bark and clove bud*	Cellulose acetate nanofibers	Electrospinning	Fresh grapes and tomatoes	The use of 50% *w*/*w* cinnamon oil (55.56% w/w cinnamaldehyde) and CBO (with 75.82% *w*/*w* eugenol)-loaded CANFs as an active food packaging membrane for a shelf-life study of fresh grapes and tomatoes at 4 °C confirmed the microbiological safety of consumption for 40 days and enhanced sensory and physicochemical properties for up to 30 days, compared to just 15 days for the controls.	[37]
*Artemisia absinthium*	Salep gum containing chitosomes (chitosan-coated essential oil-loaded nanoliposomes)—Salep-NLPs-CH	The film-forming solutions were desiccated at 35 °C for 48 h after being cast onto polystyrene plates	Toast bread	Salep–NLPs–CH film proved most effective in preserving bread color over time due to its antioxidant and antifungal properties. Mold growth was not detected until day 44, attributed to the chitosan and AEO slow release. In contrast, mold appeared earlier on samples packaged with Salep–NLPs and Salep-free AEO films. The Salep–NLPs and Salep–NLPs–CH films exhibited higher overall acceptance, likely due to the preservation of color, aroma, and texture during storage.	[38]
*Cinnamon*	CS/ Starch	Casting	Raw beef meat	Film packaging incorporated into EO with or without cellulose nanofibers had the ability to effectively reduce the bacterial load of raw beef meat samples and thereby enhance the shelf life. This property could be due to the combined effect of the chitosan in starch/chitosan/cellulose along with the CEO present in the active packaging material.	[22]
*Cinnamon*	Gelatin/pullulan	The film-forming solution was poured onto a Teflon-coated glass plate and dried at room temperature for 60 h	Meat	The low pH change indicated that meat packaged with active gelatin/pullulan-based composite films incorporated with cinnamon essential oil-loaded metal–organic frameworks can inhibit food quality deterioration after 16 days. The active film maintained the microbial load low. The results show that the incorporation of cinnamon essential oil-loaded metal–organic frameworks helped to prolong the shelf life of beef.	[21]
*Alpinia galanga*	PVA-acetylated pullulan polysaccharides	Casting	Chicken meat	Alpinia galanga essential oil components in the composite plastic provided favorable inhibiting the oxidation of proteins and lipids during shelf-life and inhibitory effects against *E. coli* and *S. aureus.*	[19]
*Melissa officinalis* L	Carboxymethyl chitosan, and locust bean gum	Coating	Large yellow croakers (Pseudosciaena crocea)	Ultrasound treatment (US) and a bioactive coating (CMCS), alone and combined, significantly inhibited microbial growth and lipid oxidation in yellow croakers during cold storage. The US+CMCS treatment was the most effective, extending shelf life considerably, compared to control and individual treatments.	[108]
*Kiwifruit seed*	Sodium alginate	Film by casting and coating material	Persimmon fruit	Applying the coating material to persimmon fruit resulted in reduced weight loss and helped maintain both firmness and respiration rate. The antifungal properties of the coating were further enhanced by the addition of EO.	[20]
Lavender flowers	Polyvinylpyrrolidone (PVP)	Centrifugal spinning	Minced lamb meat	Over the storage period, 1%, 3.5%, and 7% LEO nanofiber mats effectively suppressed meat oxidation. Microbial counts remained below acceptable limits for all samples (except for the 1% LEO PVP sample). After 5 days, yeast and mold count in the 3.5% and 7% LEO-containing samples were lower than their initial levels, likely from the intense and controlled release of LEO. The positive antimicrobial effect of the PVP film is attributable to the LEO’s activity against aerobic bacteria.	[97]
Clove	PLA; polylactic acid/modified thermoplastic starch (TPS)	Hot pressing	Shrimp	Shrimp packaged in polylactic acid incorporated with clove essential oil film (PC) maintained better quality than control (without EO). PC-packaged shrimp stayed below spoilage pH and microbial limits until day 10, compared to the control. The PC film, with its dense, crystalline structure, facilitated the controlled release of clove EO, extending shrimp shelf life.	[96]
Tea tree	CS	Solution was poured into a film-forming container and dried	Fresh cut pork	The soybean separation protein (SPI)–carboxymethyl cellulose (CMC) emulsion (SCCE) containing tea tree essential oil (TTO) incorporated into CS matrix controlled the slow release of antibacterial and antioxidant TTO into the packaging microenvironments, prolonging the pork shelf life by 6 days.	[39]
Oregano	PSE-like chicken protein isolate (PPI)	The film-forming solution was poured into polyethylene and dried in the oven	Fresh pork	Oregano EO-loaded nanoemulsion PPI films were tested for antibacterial activity against *E. coli* and *S. aureus* using the disk diffusion method. However, no significant effect against *E. coli* was revealed. The film with 2.5% oregano EO proved the highest effective in inhibiting bacterial growth and quality deterioration in refrigerated pork, thus extending its shelf life.	[40]
Wintergreen	Dehydroabietic acid (DHA) modified chitosan	The film-forming solution was poured into the mold to form a film	Mandarin oranges	Films incorporated into EO were able to delay the loss of antioxidant activity, improve the antifungal property against penicillium and prolonging the shelf life of mandarins up to 18 days.	[109]

Legend: EO—essential oil; CS—chitosan; PLA—polylactic acid; PVA—poly(vinyl alcohol); PEG—poly(ethylene glycol); PVP—polyvinylpyrrolidone; CFU—colony forming unit.

**Table 5 foods-14-02713-t005:** Studies with active packaging examples using natural extracts as active components.

Natural Extract	Polymer	Incorporation Method	Food Matrix	Main Results	Ref.
Green tea extract	Food contact polyamide (Nylon 6)	Adsorption technique	Fresh minced beef	The active film incorporated with green tea extract presented excellent antioxidant capacity. The polyamide exhibited good film-forming properties with green tea extract incorporated. The active film protected the beef’s color, as well as its lipid oxidation and variation in metmyoglobin values up to 23 days at 4 °C.	[103]
Chinese hawthorn fruit extract	Chitosan–gelatin blend film	Casting method	-	Chinese hawthorn fruit extract was successfully incorporated in a chitosan–gelatin blend film. The active film presented significantly improved mechanical and water vapor barrier properties. The addition of the extract also improved the light barrier and antioxidant properties of chitosan–gelatin films. The main polyphenols identified were epicatechin, chlorogenic acid, and procyanidin B2.	[110]
Mango leaf extract	Chitosan	Casting method	Cashew nuts	The addition of mango leaf extract increased the film’s antioxidant activity, thickness, opacity, tensile strength, and surface hydrophobicity. On the contrary, it reduced the water vapor permeability, water solubility, and elongation at break. The active film was able to protect cashew nuts from oxidation for 28 days, compared to commercial films.	[83]
Silver nanoparticle extract of *Artemisia scoparia*	Calcium alginate	Incorporation/casting method	Strawberries and loquats	A significant enhancement was observed in the quality parameters of strawberries and loquats, including reduced acidity loss, minimized soluble solid content and weight loss, and overall quality preservation. The active coating also demonstrated high antimicrobial activities.	[111]
Poplar hot water extract	Bentonite and chitosan	Casting method	-	The active film with poplar hot water extract incorporated presented greater antioxidant properties, enhanced UV blocking properties, and improved water vapor and oxygen barrier properties. The authors concluded that the new active film is a potential sustainable food packaging material.	[112]
Garlic extract	Polyethylene, Ethylene-vinyl alcohol copolymer and zein	-	Sliced pan loaf	The new active film with garlic extract developed presented antifungal activity against *Penicillium expansum*. The active film successfully delayed fungal growth in the bread, compared to the control, during the 30 days of storage.	[104]
Chitosan and rosemary extract	Poly (lactic acid)	Melt mixing	-	The incorporation of rosemary extract and chitosan on the PLA matrix resulted in a film packaging with improved elongation at break, mechanical strength, and thermal stability, as well as antibacterial and antioxidant properties. The authors concluded that this new film can be a potential active packaging with the controlled release of antimicrobial/antioxidant compounds.	[102]
Carotenoids extracts β-carotene and lycopene extracted from carrots and tomatoes and bixin extracted from annatto seeds	Poly (lactic acid)	Casting method	Sunflower oil	The incorporation of carotenoid extracts into PLA films successfully improved the shelf life of sunflower oil as it delayed its oxidation. Films with lycopene and β-carotene extracts exhibited better protection against UV light and oxygen barrier properties. Nonetheless, films with bixin extract demonstrated superior capacities in protecting sunflower oil with the best antioxidant properties.	[113]
Silver nanoparticles from *Nigella sativa* seedcake extract	Chitosan	Casting method	-	The incorporation of silver nanoparticles improved the film’s mechanical properties; specifically, it improved the film’s tensile strength and elongation and reduced the water vapor permeability. The silver nanoparticles also enhanced the film with antibacterial properties.	[114]
Rosemary extract	Whey Protein	Casting method	Salami	The whey protein film incorporated with rosemary extract was effective against the lipid oxidation of salami during 90 days at 5 °C. The active film was able to delay lipid oxidation of salami, as the samples presented lower values of MDA and hexanal in comparison with the control.	[85]
*Ginkgo biloba* extract	Gelatin	Casting method	-	The authors concluded that the incorporation of ginkgo biloba increased the films’ tensile strength and decreased their elongation at break, moisture content, solubility, and water vapor permeability. The extract also added to the film antioxidant properties and antimicrobial activity against *S. aureus* and *Candida albicans*.	[115]
*Salvadora persica* L. extract and titanium dioxide nanoparticles	Carboxymethyl cellulose	Casting method	-	The study concluded that the incorporation of Miswak (*Salvadora persica* L.) extract and titanium dioxide nanoparticles into a nanocomposite carboxymethyl cellulose film improved its properties, specifically the thermal stability, oxygen and water vapor permeability, and antimicrobial activity, namely against *E. coli* and *S. aureus*.	[116]
Oregano essential oil, rosemary extract, and green tea extract	Poly(3-hydroxybutyrate-co-3-hydroxyvalerate)	Electrospinning	-	The addition of the active compounds to the film increased its opacity and decreased its hydrophobicity. The active compounds also conferred antioxidant and antimicrobial properties to the film.	[117]
Extract from oregano waste	Carbon dioxide-derived poly (propylene carbonate) and cellulose acetate	Casting method	-	The new food packaging developed presented good mechanical, thermal, and water vapor barrier properties. The addition of oregano waste extract also improved the film’s antioxidant and antimicrobial properties. The authors also concluded that the film is biodegradable, posing as a candidate for a new sustainable active food packaging material.	[118]
* Rheum ribes * L. Extract	Methylcellulose polymer	Casting method	-	The authors found that the addition of the extract increased the thickness and opacity of the film while decreasing solubility and water vapor permeability. In addition, it enhanced the tensile strength of the films but reduced their elongation at break. The active film presented increased antioxidant capacity and antimicrobial activity.	[119]
Banana peel extract	Chitosan	Casting method	Apple fruit	The chitosan film incorporated with banana peel extract presented increased thickness, opacity, and tensile strength but reduced elongation at break, solubility, and water vapor permeability. The addition of the extract also improved the film’s antioxidant properties. The active coating was applied to apples and demonstrated to increase their shelf life, as those apples presented lower respiration rates, weight loss, and soluble solid content, as well as higher firmness, titratable acidity, and ascorbic acid content, compared to the control.	[120]
Propolis extract	Poly lactic acid	Casting method	Meat sausage	Incorporating propolis ethanolic extract into PLA films can increase the films’ thickness and opacity but decrease the tensile strength and elongation at break. When applied to sausage slices, the active film presented enhanced antioxidant activity and antimicrobial activity.	[121]
* Aloe Vera * skin extract	Poly(ethylene oxide)	Electrospinning technique	-	The new active film was evaluated by different assays that concluded that the aloe vera skin extract was successfully incorporated into the poly(ethylene oxide), obtaining a smooth, defect-free, non-woven, and self-standing film. The active film presented a slight reduction in the thermal stability but increased antioxidant activity.	[122]
Aqueous rice straw extract	Native potato starch	Melt blending and compression molding techniques	-	The incorporation of rice straw extract significantly enhanced the films’ antioxidant activity, slightly modified the mechanical strength, and also improved the films’ barrier properties by reducing water vapor permeability. The authors concluded that the films could help protect food from moisture loss and oxidation, thus extending the shelf life of food products.	[123]
Cocoa bean shell extract and zinc oxide nanoparticles	Pectin-based films	Casting method	-	Zinc oxide nanoparticles and cocoa bean shell extract addition to the pectin-based film improved the oxygen, thermal, and UV barrier properties of the films, with the oxygen barrier improved by 50% and the screen to UV radiation reaching 98%.	[82]
Olive leaf extract	Carrageenan	Casting method	Lamb meat	The addition of the extract to the film led to increased opacity and altered mechanical strength but maintained adequate flexibility and barrier properties suitable for packaging applications. The film also presented antimicrobial activity against *E. coli*. The active film was able to preserve lamb meat under refrigeration, as it significantly inhibited microbial growth, helped stabilize the pH of the meat, prevented the oxidation of lipids and proteins, and improved the sensory attributes, given that it helped maintain the red color of the meat, reduce discoloration, and retain moisture, thereby preventing excessive drying and texture deterioration.	[124]
Longan ( * Dimocarpus longan * ) peels	Paper from rice straw fibers	Casting method	-	The authors developed paper with adequate strength and flexibility to be used as packaging material with good moisture resistance and strong antimicrobial activity against both *S. aureus* and *E. coli* bacteria.	[125]
Mangosteen peel extract and zinc oxide nanoparticles	Soy protein isolate	Casting method	-	The addition of the active compounds significantly improved the film’s mechanical properties, as an increased tensile strength and elongation at break was observed. In addition, the active film presented excellent UV-blocking properties. The active films exhibited notable antioxidant activity and effective antibacterial activity against common foodborne pathogens, including *E. coli* and *S. aureus*.	[126]
* Nervilia fordii * extracts	Poly (vinyl alcohol) and polyvinyl (pyrrolidone)	Electrospinning technique	Encapsulated fish oil	The researchers were able to develop a film with uniform diameters and smooth surfaces. The addition of the extract enhanced the tensile strength and flexibility of the films and attributed a significant antioxidant capacity.	[127]
Pineapple peel extract	Poly (vinyl alcohol)–corn starch	Casting method	-	The incorporation of the extract decreased the transparency and the tensile strength of the film but increased the elongation at break and the water vapor permeability. The active film presented improved thermal stability and significantly enhanced antioxidant activity.	[128]
Chinese cinnamon (*Cinnamomum cassia*) extract	Whey protein concentrate	-	Eastern European curd cheese	The study demonstrated that the edible coating could efficiently prolong the shelf life of perishable curd cheese as it successfully inhibited microbial growth. Sensory evaluations (odor, taste, texture, appearance, and overall acceptability) indicated that the active coating did not negatively affect the cheese’s overall acceptability.	[129]
Blueberry, red grape, and parsley by-product extracts	Chitosan	Casting method	-	The addition of the extracts increased the water vapor transmission rate but decreased oxygen permeability. The swelling degree decreased with higher concentrations of extracts, indicating improved structural integrity. Both the antioxidant and antimicrobial activity of the films was enhanced by the incorporation of plant extracts.	[130]
Red cabbage (*Brassica oleracea*), sweet potato (*Ipomoea batatas*), and blue tea (*Clitoria ternatea*) extracts	Carrageenan and chitosan	Casting method	Freshly cut apple pieces	The incorporation of different extracts into carrageenan-based films resulted in films with higher mechanical strength, total polyphenol content, and antioxidant activity. In addition, when applied to freshly cut apples, the films presented reduced browning intensity and improved antioxidant activity compared to the control.	[131]
Beetroot peel extract	Gelatin–sodium alginate	Casting method	Beef meat	The study concluded that the inclusion of beetroot peel extract significantly improved the total phenolic content and consequently the antioxidant capacity. The active film was robust and flexible, demonstrating good tensile strength and elongation at break. In addition, the active film presented a reduction in water vapor permeability. The active film successfully increased the minced beef meat shelf life as it led to a reduction in thiobarbituric acid reactive substances values and inhibited microbial growth.	[80]
Pitanga leaf hydroethanolic extract and/or nisin	Gelatin	Mechanical spreading technique	Sliced dried-cured coppa	The authors concluded that the bi-layer active film effectively maintained the quality and sensory properties of the meat during storage. The active film reduced moisture loss, which maintained the texture and prevented excessive drying of the meat. The active film inhibited lipid oxidation and microbial growth during storage, extending the shelf life of the coppa slices. In addition, the active film helped retain the characteristic flavor and aroma of the coppa and maintain a more stable color profile during storage.	[132]
Jaboticaba peel extract	Carrageenan	Casting method	-	The jaboticaba peel extract presented excellent antioxidant and antimicrobial properties. The incorporation of the extract into the carrageenan matrix increased the film’s thickness and Young’s modulus and decreased the elongation capacity, tensile strength, water vapor permeability, and swelling. Nonetheless, the extract improved the opacity of the film, giving it UV–vis light barrier properties.	[133]
Watermelon rind extract	Polyvinyl alcohol, corn starch, glycerol	Casting method	Freshly cut purple cabbage	The addition of the watermelon rind extract to the composite film improved the barrier, antioxidant, and antimicrobial properties of the film. The active film was able to significantly reduce the microbial count of freshly cut purple cabbage, and it did not affect its sensory attributes.	[84]
Peony leaf extract	Chitosan	Casting method	Apples	The incorporation of peony leaf extract into chitosan film improved the film’s water vapor permeability, thermal stability, and opacity, but it negatively influenced the packaging appearance. Nonetheless, it presented as a good UV and light protector of the packed food. The active packaging was effective in retarding the natural browning process of fresh apples during storage.	[87]
Date palm pit extract	Alginate	Casting method	-	The active film demonstrated significant antioxidant activity. The active film presented good oxygen and grease barrier properties and a glossy appearance, and it was water-soluble and tasteless. The incorporation of date palm pit extracts improved water vapor barrier properties, tensile strength, and elongation at break.	[134]
Green tea extract	Low-density polyethylene	Extrusion process	Fresh orange juice	The study concluded that the active films with green tea extract were effective in extending the shelf life of fresh orange juice. The active packaging inhibited microbial, yeast, and mold growth for up to 14 days. The films decreased oxidation processes, with low levels of ascorbic acid degradation and the development of brown pigments, preventing the degradation of the juice’s quality over time.	[135]
Carboxylated cellulose nanocrystal and beetroot extract	Sodium alginate	External gelation method	Fresh pork external fat	The active film presented improved mechanical and antioxidant properties. The active compound enhanced the ability to block UV light and functioned as a real-time freshness indicator by changing color when spoilage thresholds were exceeded during storage.	[136]
*Portulaca oleracea* extract	Chitosan–starch	Casting method	Chilled pork meat	The developed active film presented excellent antioxidant capacity, good water barrier properties, and mechanical strength. The film was applied to chilled pork meat and was able to delay the lipid oxidation and meat spoilage; in addition, it protected the meat’s color during storage.	[88]
*Zanthoxylum bungeanum* leaf extract	Soy protein isolate	Casting method	Cherry tomatoes	The incorporation of the extract improved the films’ tensile strength, water barrier properties, UV-light blocking properties, and antioxidant activities. When applied to cherry tomatoes, the active film effectively maintained the quality of the tomatoes during storage, reducing weight loss and delaying spoilage compared to control.	[137]
*Ficus racemosa* fruit extract	Chitosan and sodium alginate	Casting method	-	The extract was successfully incorporated into the chitosan–sodium alginate matrix, originating a uniform and smooth surface and an improvement of the thermal stability of the films. The active films exhibited enhanced antioxidant activity with the incorporation of the *F. racemosa* extract.	[138]
Olive pomace extract	Poly lactic acid and polypropylene	-	Freshly cut Royal Gala apples	The natural extract reduced the growth of mesophilic bacteria and filamentous fungi for at least five days and inhibited the growth of coliforms for up to 12 days. The extract increased the antioxidant activity of the fruits without significant changes in their firmness and preserved their color after the initial browning of the samples.	[139]
* Viola odorata * flower extract	Potato starch	Casting method	Chicken filets	The incorporation of the extract into the film improved its phenolic content, antioxidant capacity, and antibacterial efficacy against common foodborne pathogens, including *E. coli*, *S. aureus*, and *Salmonella typhimurium*. The active film presented good light-blocking activity, especially against UV waves and improved permeability to water vapor. The active films effectively inhibited lipid oxidation and microbial growth in the chicken filets, thereby extending their shelf life compared to control samples.	[140]
Garlic extract	Chitosan–starch	-	Green and yellow bell peppers	The chitosan–starch garlic extract film demonstrated its potential as food packaging as it protected the bell peppers from bacterial growth and weight loss, protecting their general appearance during storage.	[86]
Propolis extract	*Lepidium sativum* seed mucilage	-	Buffalo meat	The active coating developed exhibited significant antioxidant and antimicrobial properties. The active coating was able to reduce lipid oxidation and microbial growth in buffalo meat during storage. The active coating was also able to minimize weight and texture losses during display and enhance the overall acceptability of the meat.	[141]
Forsythia flower extract	Starch–montmorillonite	Solution flow delay method	Cherry tomatoes	The authors concluded that the incorporation of the extract improved the film’s antioxidant and UV protection properties, as well as its thermal stability. When applied to fresh tomatoes, the active film preserved firmness, minimized nutrient loss, boosted vitamin C content, reduced decay rates, and consequently prolonged the tomatoes’ shelf life.	[142]
Olive pomace, grape marc, and moringa leaves extracts	Cellulose	-	Ground beef	The packaging material’s antioxidant qualities were significantly enhanced by the use of natural extracts, in addition to successfully decreased lipid peroxidation in food products. Additionally, over a 16-day period, the active packaging reduced lipid oxidation by at least 50% when applied to ground beef.	[143]
Apricot kernel seed extract	Chitosan	Casting method	-	The authors concluded that the incorporation of apricot kernel seed extract into the chitosan matrix significantly improves its mechanical strength, thermal stability, and barrier properties. Additionally, the active films exhibit enhanced antioxidant and antimicrobial activities, which are crucial for extending the shelf life and ensuring the safety of packaged food products.	[144]
* Camellia sinensis * leaf extract	Pectin	Casting method	-	The addition of the extract to the pectin-based film significantly improved its antioxidant activity due to the high polyphenol content in green tea. The active films demonstrated improved water resistance, reducing the permeability of moisture and oxygen. However, a slight reduction in film strength was observed with higher extract concentrations, but the overall flexibility and integrity remained within an acceptable range.	[145]
* Eucalyptus citriodora * leaf extract	Chitosan/ polyvinylpyrrolidone	Casting method	-	The authors concluded that the incorporation of the extract into the films effectively inhibited microbial growth and improved the mechanical properties by making them more robust and durable. The extract improved the films’ tensile modulus, yield strength, and tension at the break.	[146]
Blueberry anthocyanin extract	Wheat gluten protein and apple pectin	Casting method	Shrimp	The active film exhibited a uniform and compact structure after incorporation of the extract, effective water vapor permeability, and improved mechanical strength. The addition of the extract improved the antioxidant activity, which can help in delaying lipid oxidation in food products. When used to monitor shrimp spoilage, films changed color with volatile amine release, visually indicating freshness over 18 days.	[147]
* Hibiscus sabdariffa * l. extract	Potato starch and polyvinyl alcohol	-	-	The active film presented significant antioxidant capacity and antibacterial activity against common foodborne pathogens, including *E. coli* and *S. aureus*. The active film presented improved the tensile strength and flexibility and reduced water vapor permeability.	[148]

**Table 6 foods-14-02713-t006:** Studies with active packaging examples using individual compounds from essential oils or natural extracts as active components.

Compound	Polymer	Packaging Preparation Method	Food Matrix (If Applied)	Main Results	Ref.
Cinnamaldehyde (CIN)	Ethylene vinyl alcohol (EVOH)	Solvent casting and melt extrusion	NA	The study successfully developed bioactive EVOH films containing 1, 3, and 5% of cinnamaldehyde using a hybrid solvent-casting and melt-extrusion method. The films exhibited antioxidant activity, UV-blocking properties, and antifungal properties against *Penicillium expansum*. The incorporation of cinnamaldehyde also improved the films’ flexibility and transparency while maintaining their mechanical integrity, making them suitable for industrial-scale food packaging applications.	[94]
Cinnamaldehyde, carvacrol, and eugenol	Zein, polyethylene glycol (as a hydrophilic plasticizer), oleic acid (as a hydrophobic plasticizer)	Solvent casting	NA	The study successfully developed biodegradable zein-based films incorporated with cinnamaldehyde, carvacrol, and eugenol with antimicrobial properties, with cinnamaldehyde showing the strongest activity, particularly against *S. aureus* for a period up to 96 h. However, while higher cinnamaldehyde concentrations improved film flexibility, they reduced tensile strength, with PEG proving more effective than OA as a plasticizer for enhancing mechanical properties. The results highlight the potential of these zein–PEG films containing 5% cinnamaldehyde as sustainable, antimicrobial packaging material.	[95]
Thymol, eugenol, and cinnamaldehyde	Pullulan	Lipid nanoparticle encapsulation	NA	This study investigated the release kinetics of thymol, eugenol, and cinnamaldehyde from pullulan-based biodegradable films, comparing liquid-lipid nanoparticles and solid-lipid nanoparticles as carriers. The results showed that solid-lipid nanoparticles films provide a faster release of antimicrobials due to the expulsion of active compounds during lipid crystallization. Among the tested compounds, thymol exhibits the highest release rate.	[149]
Cinnamaldehyde	Chitosan and acidified montmorillonite (MMT)	Solvent casting	NA	In this study, chitosan-based films with acidified MMT loaded with cinnamaldehyde were developed and evaluated. The films exhibited improved mechanical strength, UV resistance, and prolonged cinnamaldehyde release (using isooctane as a fatty food simulant), along with significant inhibition of *S. aureus* and *E. coli*.	[150]
Carvacrol, thymol, and cinnamaldehyde	Low-density polyethylene (LDPE) and chitosan	Cocrystallization and Solvent casting	White grapes	This study developed a cocrystal-based active packaging using carvacrol, thymol, and cinnamaldehyde anchored to a chitosan-coated LDPE. The active films showed antimicrobial activity against *E. coli*, *Salmonella Typhimurium*, *S. aureus*, and MR *S. aureus*. The films extended the shelf-life of white grapes by 7 days, maintaining their sensory quality.	[151]
Cinnamaldehyde	Chitin from shrimp and polyvinyl alcohol (PVA)	Solution casting method	Cherry tomatoes	Active packaging films by incorporating β-cyclodextrin/CIN inclusion complexes into chitin/PVA blends using a solution casting method. The films demonstrated a sustained release of CIN, enhanced antimicrobial activity against *S. aureus*, *Bacillus subtilis*, *S. typhimurium*, *Aspergillus niger*, *Aspergillus flavus*, and *Penicillium citrinum*. The film with 3% β-cyclodextrin/CIN effectively preserved the cherry tomatoes by reducing weight loss, maintaining hardness, and inhibiting microbial growth over 10 days.	[152]
Cinnamaldehyde	Chitosan and dialdehyde carboxymethyl cellulose (DCMC)	Solution casting method	Strawberries	Active packaging chitosan/DCMC-based films with zein nanoparticles loaded with CIN were successfully produced. The films exhibited improvements in mechanical strength, water vapor and oxygen barrier properties and UV-blocking ability. The film with 35% zein nanoparticles and CIN effectively preserved the strawberries by reducing microbial growth, weight loss, and maintaining quality over 7 days of storage.	[153]
Thymol, carvacrol, limonene and cinnamaldehyde	Polylactic acid (PLA)	Solvent-casting technique	NA	This study developed a PLA-based active film incorporating thymol, carvacrol, limonene, or CIN using solvent casting. The films exhibited significant antioxidant activity, with carvacrol demonstrating superior performance compared to the other individual compounds and their triple blends. Regarding the films’ characteristics, the active films exhibited plasticization effects and maintained PLA’s crystallinity, with CIN slightly reducing thermal stability.	[154]
Thymol	Polycaprolactone (PCL)	Solution blow spinning	NA	Thymol was encapsulated in covalent organic frameworks and incorporated in PCL through solution blow spinning. The films exhibited controlled thymol release and antimicrobial activity against *S. aureus* and *E. coli*.	[155]
Thymol	LDPE	Melt extrusion process	Fresh “scaloppini”-type pork meat filets	Halloysite nanotubes were impregnated with thymol to form a hybrid nanostructure. Then, this nanostructure was incorporated in LDPE at 5, 10, and 15%. The active films exhibit significant antioxidant activity and improved barrier properties. The LDPE with 10% of the thymol nanostructure optimally preserved pork meat by significantly reducing lipid oxidation. The research also established a linear correlation between TBARS and heme iron measurements, offering a faster method to assess meat spoilage.	[156]
D-Limonene	PVA	Electrospinning, with additional ultrasonic processing	NA	PVA/D-limonene composite fibers, with the optimized ratio of 7:3 and an ultrasonic processing time of 15 min, were developed. The active composite fibers presented antimicrobial activity against *E. coli* and *S. aureus* and enhanced degradability and homogeneity due to ultrasonic treatment.	[157]
D-Limonene	Sodium alginate	Phase inversion and brush-coated application	Bananas (*Musa sapientum* Linn.)	The study developed an edible coating using a sodium alginate D-limonene nanoemulsion, which demonstrated antimicrobial activity against *S. aureus*, *L. monocytogenes*, *Salmonella enterica*, and *E. coli*, and effectively extended the shelf life of bananas by reducing weight loss and delaying ripening. The 1.0% D-limonene concentration was optimal, while higher concentrations caused undesirable visual effects.	[158]
Eugenol	Poly(3-hydroxybutyrate-co-3-hydroxyvalerate) (PHBV)	Electrospinning	NA	The authors developed multilayer food packaging using electrospun PHBV fibers loaded with eugenol, which demonstrated antimicrobial activity against *S. aureus* and *E. coli*. The optimized 15 wt.% eugenol–PHBV monolayer, exhibited enhanced barrier properties, hydrophobicity, and thermal stability.	[159]
Catechin	Zein reinforced with cellulose nanocrystals	Solution casting	Soybean oil	An antioxidant-active nanocomposite film using zein, catechin/β-cyclodextrin nanoparticles, and cellulose nanocrystals was created in this study. The active film exhibited enhanced mechanical strength, barrier properties, and oxidative stability. Also, the film effectively inhibited lipid oxidation in soybean oil, demonstrating potential for active food packaging applications.	[160]
Catechin	Octenyl succinic anhydride starch and Pullulan	Electrospinning and glutaraldehyde vapor phase crosslinking	Strawberries	In this study, a starch-based nanofiber film was developed using electrospinning and glutaraldehyde crosslinking, incorporated with catechin. The active film presented antimicrobial activity against *S. aureus* and *E. coli* along with in vitro antioxidant activity. The active film successfully extended the strawberries’ shelf life, maintaining their freshness by 6 days.	[161]
Linalool and thymol	Polyethylene	Direct mixing and compression molding	Mozzarella cheese	The study developed antimicrobial polyethylene films infused with linalool or thymol, which significantly inhibited *S. aureus* and *E. coli* growth in mozzarella cheese, extending its shelf life. Thymol (2%) was most effective, preventing bacterial contamination and reducing mold/yeast proliferation.	[162]
Thymol and carvacrol	PLA	Melt-processed by injection molding	Blackberries and raspberries	This study evaluated a PLA packaging containing thymol or carvacrol complexed with β-cyclodextrins for preserving blackberries and raspberries during cold storage. The active packaging showed significant antioxidant and antimicrobial properties, improving fruit quality and extending shelf life by one week over commercial packaging. Sensory evaluation confirmed no negative impact on flavor or aroma, supporting the potential of these natural compounds for food preservation.	[163]

Legend: CIN—cinnamaldehyde; NA—non-applicable; EVOH—ethylene vinyl alcohol; PEG—polyethylene glycol; MMT—montmorillonite; LDPE—low-density polyethylene; PVA—polyvinyl alcohol; DCMC—dialdehyde carboxymethyl cellulose; PLA—polylactic acid; PCL—Polycaprolactone; PHBV—poly(3-hydroxybutyrate-co-3-hydroxyvalerate).

## Data Availability

No new data were created or analyzed in this study. Data sharing is not applicable to this article.

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
