# Peer review of "Emerging Trends in Active Packaging for Food: A Six-Year Review"

_foods, 2025, doi:10.3390/foods14152713_

Round 1

Reviewer 1 Report

Comments and Suggestions for Authors

Emerging Trends in Emission Active Packaging for Food: 2 A Six-Year Review

ABSTRACT:

  • The phrase “emission-type active packaging” might be clarified for broader audience understanding (e.g., “release-type active packaging”).
  • Consider specifying examples of “commercial applications” briefly in the abstract to increase impact.
  • Suggest tightening the sentence structure for smoother reading. For example:
    Original: "These compounds, derived primarily from plant-based sources, exhibit significant antioxidant and antimicrobial activities, contributing to extended shelf life and enhanced food quality."
    Suggested: "Primarily plant-derived, these compounds exhibit significant antioxidant and antimicrobial activities, extending shelf life and enhancing food quality."
  • “Circular economy” is included as a keyword but not explicitly discussed in the provided text. Consider integrating this concept more clearly or removing if not central.

INTRODUCTION:

  • The introduction clearly explains the need for active packaging and the shift from synthetic to natural additives. However, some details still need to be addressed.
  • Line 58–60: The transition from synthetic to natural additives is abrupt. Consider adding a sentence to bridge why industry is shifting specifically to phenolics and EOs?
  • The Term EO has been defined in the line # 73 but afterwards sometimes “EOs” is used, sometimes spelled out as “essential oils.” Ensure consistent use after first definition.
  • (Lines 98–112): The discussion of publication trends (PubMed/Web of Science) is informative but would be greatly enhanced with a graphical representation (e.g., bar chart or pie chart).

Active Compounds:

  • Consider standardizing terminology. At times, the manuscript refers to "EOs," "essential oils," and “volatile oils” interchangeably. Define and stick with a consistent term.
  • Although mostly clear, some sentences could be more concise. For example:

"Essential oils hold great promise as such additives for these films and coatings, to offer a natural approach to controlling microbial growth and preserving food quality."

could be simplified to:
"Essential oils offer a natural solution for microbial control and food preservation in bio-based films

  • While the review promises to highlight trends from 2019–2024, the analysis of those trends is descriptive rather than critical. A more systematic or thematic structure (e.g., EO by plant source, application area, or packaging type) would enhance clarity and originality.
  • There is no clear synthesis of which specific trends (e.g., nanoencapsulation, synergistic EO blends, biopolymer-EO interactions) are emerging. The manuscript would benefit from clearer sub-sectional organization under Section 2.1.4, focused on specific innovations.
  • Some concepts are repeated multiple times, such as the general benefits of EOs (antioxidant, antimicrobial). For example, the roles of terpenes and phenolic compounds are explained more than once in slightly different ways. This could be streamlined to improve conciseness and flow.
  • Section 2.1.3 would benefit from deeper mechanistic insight into how specific EO components interact with microbial membranes or oxidative systems. Mentioning specific modes of action (e.g., membrane disruption, ROS generation) would elevate the scientific rigor.
  • To consider better language composition, avoid using the same words consistently. For example

Under subheading 2.1.5… the two initial paragraphs start with However… one must be replaced with another grammatical structure.

  • Moreover, this section describes limitations like strong aroma and volatility, but fails to critically point out the challenges in EO-food compatibility, migration rate control, regulatory hurdles, or consumer perception. These are vital for a comprehensive review and should be discussed more robustly.
  • A separate subsection discussing regulatory, economic, and consumer acceptance challenges is strongly recommended.
  • Some general statements (e.g., “EOs represent a safe and biodegradable alternative...”) require stronger referencing, ideally from meta-analyses or systematic reviews. It is also unclear whether Web of Science and PubMed search methodologies followed PRISMA or other standard frameworks.
  • The manuscript briefly touches on synergistic effects and structure-activity relationships but falls short of explaining how EO constituents (e.g., phenolics, terpenes) interact with microbial membranes or oxidative agents.
  • Adding figures or diagrams summarizing mechanistic pathways would enhance comprehension
  • Table 1 heading “Publication data” should be “Year of publication”.

Section 2.2.4:

  • The section cites many relevant studies but presents them in a largely descriptive, study-by-study manner without synthesizing common findings, limitations, or gaps.
  • Create a brief summary paragraph after the listed examples to draw general insights (e.g., polymer combinations that work best, extract types most effective against specific spoilage mechanisms, or packaging performance trade-offs, among others).
  • The mention of circular economy and food waste valorization is important but underdeveloped.
  • Expand this point with a couple of concrete examples (e.g., extract derived from fruit peels or vegetable pomace used in food packaging), along with a sentence on the potential for scalability.
  • While greener extraction techniques are mentioned, their comparative environmental benefits are not quantified or well contextualized.
  • Consider adding brief comparisons (e.g., solvent use, energy consumption) or referencing LCA (Life Cycle Assessment) studies that validate the “green” claims.
  • Several times, the text notes that “additional applications are discussed in Section 4” without giving a hint of what specific applications will follow.
  • Include a more informative transition e.g., “Further applications, including meat and dairy preservation using grape seed extract-based films, are detailed in Section 4”.
  • Figure 2: The figure caption is very brief and lacks a proper explanation in the main text.
  • Expand the caption to indicate the advantages of each method class and provide one or two examples in the text (e.g., “MAE has been shown to yield up to 30% more phenolic compounds compared to Soxhlet extraction…”).

Biopolymers:

  • The section describes the advantages and limitations of several biopolymers (e.g., PLA, chitosan, starch) but does so in a somewhat superficial and repetitive manner.
  • Rather than listing limitations and potential improvements polymer-by-polymer, synthesize the information by comparing them directly (e.g., a brief comparative paragraph), highlighting which polymers are best suited for which applications.
  • While the conclusion acknowledges difficulties in scaling up, the main text lacks technical detail on the processing and compatibility challenges faced when incorporating essential oils (EOs) into polymer matrices.
  • Expand on the thermal, rheological, or structural challenges in film formation with active components, especially for polymers like PLA or chitosan.

Application of New Active Food Packaging to Foods:

  • There is an overreliance on descriptive literature summaries in Section 4.
  • The examples of packaging applied to food (meat, cheese, fruits) are informative but lack a critical synthesis of common findings or performance benchmarks.
  • After the case studies, include a summary comparing key performance indicators (e.g., shelf-life extension days, microbial load reduction, sensory scores), and discuss which combinations of biopolymer + EO/extract were most effective.
  • There are abrupt transitions (e.g., from Section 3 to Section 4) and duplicated or awkward phrases like “TablesTable 3” (line#459)
  • Smooth transitions between sections and remove typographical errors. Clarify cross-references to tables and ensure the tables are well-integrated into the discussion.

Conclusions:

  • The conclusion covers key themes well but remains general in places. Phrases like “continued research will be critical” or “interdisciplinary collaboration is needed” are valid but vague.
  • Include more concrete forward-looking strategies—e.g., developing open-source databases for active packaging formulations, standardizing migration testing, or proposing a roadmap for regulatory harmonization.
  • The discussion does not sufficiently address the interactions between active components and the food matrix, such as migration rates, sensory thresholds, and compound stability.
  • Consider including a paragraph in the conclusion or discussion on how food matrix characteristics (fat content, moisture, pH) can influence the effectiveness of bioactive films.
  • The manuscript refers to figures and tables but lacks discussion of what the reader should observe or conclude from them.
  • Briefly describe the most important takeaway from each cited table/figure in the main text.
  • The benefits of EO incorporation (controlled release, antimicrobial activity, sensory masking) are repeated across multiple sections without new context.
  • Consolidate discussions of EO functionality and avoid redundancy.

General Comments:

  • Reduce redundancy and improve analytical synthesis.
  • Clarify transitions and correct formatting inconsistencies.
  • Add comparative summaries to enhance readability.
  • Strength en the technical depth regarding processing, performance, and food-polymer interactions.
  • Improve language by better sentence composition and reducing grammatic errors.

Comments on the Quality of English Language

The whole manuscript should be revised for the language errors. To cite a few;

Some sentences are too long or use passive voice excessively, which reduces clarity.

Example: “CO proved most effective in inhibiting metmyoglobin formation and lipid oxidation…” — consider rephrasing to make subject-action-object relationships clearer.

Duplication in Table 1:

  • Patent CN114249926B is listed twice.

Proofread for formatting consistency and clarity

Typografical errors: like;

Table 1: In the JP patent description: "...for extending the shelf life of perishable tems" → should be "items."

Author Response

Dear Reviewer #1,

First of all, we would like to sincerely thank you for your valuable comments and suggestions, which have significantly contributed to improving the quality of our manuscript (ID: foods-3734576), entitled “Emerging Trends in Emission Active Packaging for Food: A Six-Year Review.”

In response to your feedback, we have carefully revised the manuscript to address all the points you raised. For your convenience, all changes have been highlighted in yellow in the revised version of the manuscript.

Please find our detailed, point-by-point responses to each of your comments in the attached PDF.

Sincerely,
Mariana A. Andrade

Reviewer 2 Report

Comments and Suggestions for Authors

The manuscript intends to be a short review on a specific type of active food-packaging materials. The paper is reasonable written and accurate from a references point of view. The overall organization is a little bit confusing, since in every paragraph there is a mix of information on active additives and the materials used for their incorporation.

So, my suggestion is to order the discussion, starting with the paragraph dedicated to the biopolymers, then the paragraph on the active additives and finally the last paragraph. Regards to this, the tile of paragraph 4. :” Application of New Active Food Packaging to Foods” sounds strange to me. I would propose: “Examples of application of new active food-packaging materials”

Table 3 is to detailed (see column “Main results”). If authors want to comment some of the examples reported in Table 3 and their main results, I suggest to introduce them into the last paragraph. The main results reported for each example in Table 3 need to be reduced to a sentence.

Last comment concerns the Title: I suggest to delete the word “emission” and change it into: “Emerging Trends in Emission Active Packaging for Food: 2 A Six-Year Review”

Author Response

Dear Reviewer #2,

First of all, we would like to sincerely thank you for your valuable comments and suggestions, which have significantly contributed to improving the quality of our manuscript (ID: foods-3734576), entitled “Emerging Trends in Emission Active Packaging for Food: A Six-Year Review.”

In response to your feedback, we have carefully revised the manuscript to address all the points you raised. For your convenience, all changes have been highlighted in yellow in the revised version of the manuscript.

Below, please find our detailed, point-by-point responses to each of your comments in the attached PDF.

Sincerely, 
Mariana A. Andrade

Round 2

Reviewer 1 Report

Comments and Suggestions for Authors

The manuscript has been meticulously corrected and fit for publication in my opinion.